# Photostability Enhancement of Dual-Luminophore Pressure-Sensitive Paint by Adding Antioxidants

**DOI:** 10.3390/s22239470

**Published:** 2022-12-04

**Authors:** Kazuki Uchida, Yuta Ozawa, Keisuke Asai, Taku Nonomura

**Affiliations:** Department of Aerospace Engineering, Graduate School of Engineering, Tohoku University, 6-6-01 Aramaki-Aza-Aoba, Aoba-ku, Sendai 980-8579, Miyagi, Japan

**Keywords:** dual-luminophore PSP, photodegradation, antioxidant

## Abstract

Antioxidants were applied to a dual-luminophore pressure-sensitive paint (PSP), and the effects on photodegradation caused by exposure to excitation light were studied. Three types of antioxidants that are commonly used for the photostability enhancement of polymers were added to a dual-luminophore PSP, and degradation rates and pressure/temperature sensitivities were investigated by coupon-based tests. One-hour-long aging tests were performed in a pressure chamber with a continuous excitation light source under dry air and argon atmospheres at 100 kPa and 20 °C. As a result of the aging tests, a singlet oxygen quencher type antioxidant was found to reduce the degradation rate by 91% when compared with the dual-luminophore PSP without antioxidants. This implies that singlet oxygen has a dominant role in the photodegradation mechanism of the dual-luminophore PSP.

## 1. Introductions

Pressure-sensitive paint (PSP) is a global surface pressure measurement technique based on the photochemical reactions of luminescent molecules [1,2,3,4,5]. Because PSP is a molecular-level pressure sensor, it has a significantly higher spatial resolution than electric pressure transducers. PSP is an effective technique for flow visualization and diagnostics in experimental aerodynamics due to its high spatial resolution [6,7,8]. In addition, PSP is a nonintrusive technique, and it can be applied to thin models, such as a control surface or a wing trailing edge [9,10,11].

The temperature dependence of PSP luminescence characteristics, such as intensity and lifetime, is one of the largest error sources in surface pressure measurement using PSP. Because the oxygen diffusion process in a polymer is temperature dependent, PSP has an intrinsic temperature dependence [5]. Temperature-induced error becomes significant in low-speed wind-tunnel tests where pressure variation caused by fluid phenomena is small or hypersonic wind-tunnel tests with a large temperature gradient due to aerodynamic heating [12,13,14,15].

Thus far, many methods have been proposed for the correction or compensation of temperature dependence and pressure measurement accuracy improvement. Infrared (IR) thermometry is one of the most common global temperature measurement techniques. Kammeyer et al. [16] and Mitsuo et al. [17] applied IR thermometry to the temperature correction of PSP measurement. Although IR thermometry is effective for capturing the temperature distribution over a model, there are several restrictions. For instance, window glass material is limited because ordinary glass is nontransparent to IR rays [17]. In addition, some PSPs have a high IR reflectivity, and the temperature on the wind-tunnel walls can be reflected onto the PSP coating. The present authors experience that the polymer/ceramic PSP (PC-PSP) possibly has a high reflectivity for IR rays (i.e., low emissivity).

Temperature-sensitive paint (TSP), which is based on photochemical reactions similar to that of PSP, is another method for global temperature measurement. The left and right sides of a model can be painted with PSP and TSP, respectively, in the case of a symmetric flow field [18,19,20,21,22]. Obviously, this method cannot be applied to an asymmetric model or a model with a yaw angle. Furthermore, internal structures in one side of most wind-tunnel models, including pressure tubes, are not completely identical to those of the other side, and it can be a critical error source in low-speed wind-tunnel tests.

Thus, a PSP/TSP combined measurement technique that can simultaneously measure the pressure and temperature distributions of a single surface is desirable for accurate temperature correction and pressure measurement. A dual-luminophore PSP can realize such measurements [13,23,24,25,26,27,28,29]. A dual-luminophore PSP technique adopts a mixture of a pressure-sensitive luminophore and a temperature-sensitive or insensitive luminophore as a luminescent dye. The luminophore insensitive to either pressure and temperature is called a reference luminophore and is used for correction of nonuniform illumination and model variation during a wind-tunnel run [12,14]. Hereafter, we focus on a dual-luminophore PSP with a temperature-sensitive luminophore. In a dual-luminophore PSP, luminescence values from the two types of luminophores are measured separately based on the difference of wavelengths or lifetimes, and pressure and temperature information are extracted from the two types of luminescence with different pressure/temperature sensitivities.

Although a dual-luminophore PSP can measure pressure and temperature simultaneously, photochemical interaction between the luminophores often occurs and accelerates photodegradation due to the mixing of two types of luminophores in the molecular level [27,30]. Photodegradation is a luminescence intensity reduction and sensitivity change due to the decomposition or oxidization of luminescent molecules. It often rises as a technical issue in dual-luminophore PSP measurements. Photodegradation deteriorates the pressure measurement accuracy because it causes a luminescence intensity change, even at constant pressure and temperature conditions.

Hradil et al. [23] proposed a lifetime-based simultaneous pressure and temperature measurement technique. They used a short-lived ruthenium complex as a pressure-sensitive luminophore (pressure sensor) and a long-lived thermographic phosphor as a temperature-sensitive luminophore (temperature sensor). Although their paint was photostable and did not suffer from photodegradation, the temperature sensitivity of the phosphor was low (~0.3%/°C) compared with that of the pressure sensor (~1%/°C).

Katagiri et al. [31] developed the tetranuclear Eu(III) complex ([Eu4 μ-O) (L2)10] (L2 = 2-hydroxy-4-dodecyloxybenzophenone)) as a temperature sensor with a high-temperature sensitivity of 3%/°C and almost no pressure sensitivity. Although the Eu complex was photostable when it was used alone, the mixture with PdTFPP, which is a palladium porphyrin pressure sensor, was unstable to ultraviolet (UV) excitation light. The degradation rate was only a few percent for one-hour continuous irradiation under atmospheric pressure and room temperature when it was used alone, whereas the degradation rate became ten times higher when it was combined with the pressure sensor [27]. Kameya et al. [30] proposed the dual-luminophore PSP that uses PtTFPP, a platinum porphyrin, as a pressure sensor and CdSe/ZnS, a quantum dot, as a temperature sensor. The paint had a pressure sensitivity of 0.77%/kPa and a temperature sensitivity of 0.88%/°C. They also reported the deterioration of the paint photostability compared with the sole temperature sensor.

Paints are considered to have different degradation mechanisms depending on paint formulations. Basu et al. [32] reported that the degradation of a pyrene-based PSP with a silicone resin is driven by diffusion and sublimation of pyrene at the low pressure of 10 kPa and the high temperature of 50 °C, while the effect of continuous UV excitation is negligibly small. Egami and Asai [33] investigated the photodegradation of an anodized aluminum PSP (AA-PSP) using pyrene butyric acid under dry air and argon atmospheres. Consequently, photodegradation of the PSP was suppressed by half under an argon atmosphere (i.e., without oxygen), and they indicated that approximately half of the photodegradation is driven by oxygen-related processes (i.e., oxidation). In addition to the investigation of the mechanism, they applied three types of antioxidants to the PSP and reduced photodegradation by 51% at most.

In the present study, three types of antioxidants that are widely used for photostability enhancement of polymers and paints were added to a dual-luminophore PSP using a polymer/ceramic binder, and degradation rates and pressure/temperature sensitivities were investigated by coupon-based tests. One-hour-long aging tests were performed in a pressure chamber with a continuous UV excitation light source under dry air and argon atmospheres at 100 kPa and 20 °C. The photodegradation mechanism of the dual-luminophore PSP and its reduction method are discussed based on the results of the tests.

## 2. Photodegradation and Antioxidants

In this section, the photochemical mechanism of PSP photodegradation is introduced based on the previous study of the AA-PSP using pyrene butyric acid by Egami and Asai [33]. Then, three types of antioxidants are used in the present study and those functions are described.

### 2.1. Photodegradation Mechanism

Egami and Asai [33] explained the mechanism of PSP photodegradation by applying the photodegradation mechanism of polymers and paints [34,35] to a PSP luminophore. Because the photodegradation processes of polymers are driven by photochemical reactions of organic molecules, they can be applied to complex luminophores with organic ligands, such as porphyrin. Figure 1 shows a schematic of the photodegradation processes of an organic luminophore. Here, RH, hν, and a superscript asterisk * denote a luminescent molecule, excitation energy, and an excited state, respectively. Photodegradation includes two processes called photodecomposition and photooxidation.

Photodecomposition is the dissociation of photo-excited molecules. Excited luminescent molecules emit energy as light called luminescence when PSP is excited by light (generally UV or blue light). A part of the excited molecules breaks up into smaller molecules, ions, or radicals during this process. The photodecomposition process does not require oxygen. The decomposed molecules lose their original characteristic as a luminophore. In addition, free radicals produced in the process initialize radical oxidation reactions that are described later.

Photo-oxidation is an oxidation process of luminescent molecules. The process includes nonradical and radical (auto) oxidation reactions. Nonradical oxidation is an oxidation reaction by singlet oxygen. Singlet oxygen, a type of active oxygen species, is produced by an oxygen-quenching reaction of luminescent molecules in PSP. When oxygen quenching occurs, triplet oxygen ${}^{3}O_{2}$ absorbs energy from excited luminescent molecules and becomes singlet oxygen ${}^{1}O_{2}$. One singlet oxygen molecule (${}^{1}O_{2}$) can oxidize one luminescent molecule (RH) and convert into hydroperoxide (ROOH) via the following reaction.
(1)RH+O21→ROOH
Obviously, this reaction only proceeds in the presence of oxygen.

In radical (auto) oxidations, a free radical (R•) produced in photodecomposition reacts with oxygen and the chain of oxidative reactions proceeds by the following scheme (see Figure 1).

Initiation reaction: (2)$$\begin{array}{c} \begin{array}{c} RH\to R\cdot +H\cdot \end{array} \end{array}$$

Growth reaction: (3)$$\begin{array}{c} \begin{array}{c} R\cdot +O_{2}\to ROO\cdot \end{array} \end{array}$$
(4)$$\begin{array}{c} \begin{array}{c} ROO\cdot +RH\to ROOH+R\cdot \end{array} \end{array}$$
(5)$$\begin{array}{c} \begin{array}{c} ROOH\to RO\cdot +HO\cdot \end{array} \end{array}$$
(6)$$\begin{array}{c} \begin{array}{c} 2\;ROOH\to RO\cdot +ROO\cdot +H_{2}O \end{array} \end{array}$$
(7)$$\begin{array}{c} \begin{array}{c} RO\cdot +RH\to ROH+R\cdot \end{array} \end{array}$$
(8)$$\begin{array}{c} \begin{array}{c} HO\cdot +RH\to R\cdot +H_{2}O \end{array} \end{array}$$

Termination reaction: (9)$$\begin{array}{c} R\cdot +R\cdot \\ R\cdot +ROO\cdot \to \mathrm{nonradicalspecies}\\ ROO\cdot +ROO\cdot \end{array}$$
In radical oxidation, dehydrogenation of a luminescent molecule (RH) in the photodecomposition process is an initiation reaction as shown in Equation (Equation 2). Once the chain reactions start, oxidation reactions continue even without light, as shown in Figure 1. In addition, hydroperoxide (ROOH) is also dissociated into radicals RO· and HO· by excitation light as shown in Equation (Equation 5). Because oxygen is consumed in the reaction, radical oxidation only proceeds in the presence of oxygen. The radicals produced in the photodegradation processes can be terminated by numerous different combination reactions between two radicals, and the inactive products that are produced are shown in Equation (Equation 9) [35].

### 2.2. Antioxidants

Antioxidants and photostabilizers have been used in polymers and paints for the prevention of degradation reactions and improvement of photostability [34,35,36]. In the present study, three types of antioxidants—free radical scavengers, hydroperoxide decomposers, and singlet oxygen quenchers—were added to the dual-luminophore PSP. Figure 2 shows a schematic of photostabilization by antioxidants.

Free radical scavengers (FRS) react with radical intermediates in the radical oxidation and convert them into inactive species. FRSs scavenge alkylperoxy (ROO·) and alkyloxy (RO·) radicals as shown in Figure 2. Hydroperoxide decomposers (HPD) react with hydroperoxide (ROOH) and decompose it into nonradical species. In this way, HPDs prevent the radical oxidation. Singlet oxygen quenchers (SOQ) deactivate excited singlet oxygen (${}^{1}O_{2}$) and prevent nonradical oxidation. Nickel complexes are efficient quenchers of singlet oxygen [36,37,38].

As they are other types of photostabilizers, UV absorbers and hindered amine light stabilizers (HALS) are also widely used in polymers and paints. Because UV light is necessary for PSP excitation, UV absorbers cannot be applied to PSP. HALSs are remarkably effective in polymer stabilization, while they have multiple photostabilization mechanisms and are not suitable for mechanism identification of PSP photodegradation.

## 3. Experimental Setup

In this section, the paint formulations and the procedure for the one-hour-long aging tests are described.

### 3.1. Paint Formulation

In the present study, a PC-PSP-based dual-luminophore PSP was used. A polymer/ceramic binder with low surface roughness [39] was applied as a binder. Pt(II) meso-tetrakis (pentafluorophenyl) porphine (PtTFPP) and the tetranuclear Eu(III) complex ([Eu4 μ-O) (L2)10] (L2 = 2-hydroxy-4-dodecyloxybenzophenone)) [27,31] were used as the pressure sensor and the temperature sensor, respectively. Here, 2,4,8,10-tetraoxaspiro [5.5] undecane-3,9-diylbis(2-methylpropane-2,1-diyl) bis[3-[3-(tert-butyl)-4-hydroxy-5-methylphenyl] propanoate], Didodecyl 3,3’-thiodipropionate, and nickel(II) dibutyldithiocarbamate were used as antioxidants based on a FRS, a HPD, and a SOQ, respectively.

Six types of PSP sample coupons were prepared for the aging tests by changing the dye solutions. The formulation of the binder and dye solutions are listed in Table 1 and Table 2, respectively. The binder solution was prepared by mixing 0.500 g of poly (isobutyl methacrylate) (poly(IBM)) and 6.64 g of titanium oxide particles (SJR-600M, TAYCA CORPORATION, Osaka, Japan) for every 30 mL of toluene. The solution was stirred for 24 h. The six types of dye solutions were prepared by mixing component materials with the proportions listed in Table 2. Each dye solution was denoted as Pt, Eu, Dual, +FRS, +HPD, and +SOQ. The dye solutions were stirred for an hour with an ultrasonic cleaner.

The binder solution was sprayed onto aluminum sample coupons with an airbrush, and a polymer/ceramic coating was formed as a binder. After drying the binder, the dye solutions were sprayed onto the binder with an airbrush, and six types of coupons were prepared.

Figure 3 shows luminescence spectra under the air atmosphere at 100 kPa and 20 °C for each coupon measured by a spectrometer (RF-5300PC, Shimadzu Corporation, Kyoto, Japan). Because the spectrometer head was placed at the fixed position in all the measurements, a relative comparison of intensity is applicable. Peaks of the spectra of PtTFPP (the pressure sensor) and the tetranuclear Eu(III) complex (the temperature sensor) are approximately at 651 and 617 nm, respectively. Note that the measurement accuracy of the spectrometer is 1.5 nm in wavelength.

### 3.2. PSP Calibration System

Figure 4 illustrates a schematic diagram of the PSP calibration system at Tohoku University. The aging tests were conducted by using the PSP calibration system in the present study. The system consisted of a UV light-emitting diode (LED) (IL-106, HARDsoft Microprocessor Systems, Kraków, Poland) with a peak wavelength at 400 nm for PSP excitation, a 16-bit charge-coupled device (CCD) camera (ORCA II-BT 1024, Hamamatsu Photonics K.K., Hamamatsu, Japan) as a photodetector, and a pressure chamber. The distance from the UV-LED to a sample coupon was approximately 80 cm.

A filter wheel (PTCA003-04S, Hamamatsu Photonics K.K., Hamamatsu, Japan) was attached in front of the 16-bit CCD camera, which can switch the optical filters by rotating a wheel. The optical filters for the pressure sensor and the temperature sensor were a 650 ± 20 nm band-pass filter and a 610 ± 20 nm band-pass filter, respectively.

The pressure and temperature in the chamber in which the sample coupons are installed can be adjusted arbitrarily with a pressure controller and a temperature controller.

### 3.3. Test Conditions and Procedures

The one-hour-long aging tests were conducted for the six types of sample coupons under dry air and argon atmospheres at 100 kPa and 20 °C with continuous excitation. PSP images with two wavelength ranges were acquired every minute by changing the optical filters using the filter wheel.

## 4. Results and Discussions

One-hour-long aging tests with a continuous UV light source were conducted for sample coupons with and without antioxidants, and degradation rates and pressure/temperature sensitivities were investigated.

### 4.1. Degradation Rates

Figure 5 shows the degradation rates ([%/hour]) for each dye formulation. The degradation rate *D* is defined in Equation (Equation 10).
(10)$$\begin{array}{c} D=\frac{I_{t=60}-I_{t=0}}{I_{t=0}}\times 100\;\left[\%/\mathrm{hour}\right] \end{array}$$
Here, $I_{t=60}$ and $I_{t=0}$ denote the luminescence intensity at t=0 min and t=60 min, respectively. The two wavelength ranges, 650 ± 50 nm and 610 ± 20 nm, mainly contain emissions from the pressure sensor (PtTFPP) and temperature sensors (tetranuclear Eu(III) complex), respectively. Because the sample coupons Pt and Eu do not contain tetranuclear Eu(III) complex and PtTFPP, respectively, only one wavelength range is shown.

Figure 5 shows that both pressure and temperature sensors were photostable when they were separated. The photodegradation of Pt increased under the argon atmosphere, which suggests that the photodecomposition of PtTFPP is accelerated under the argon atmosphere (i.e., without oxygen).

Mixing two luminophores accelerated the photodegradation of the tetranuclear Eu(III) complex under the air atmosphere as reported in the previous studies by Mitsuo et al. [40] and Mochizuki et al. [41]. They used PdTFPP, a palladium porphyrin, as a pressure sensor and poly-IBM-co-TFEM as a binder. Table 3 compares the present study with those two previous studies by focusing on the degradation rate of the temperature sensor. Mitsuo et al. [40] did not provide the specific value of the degradation rate. Because Mochizuki et al. [41] performed their aging test for only 30 min, the degradation rates at 30 min are shown in the table.

In all dual-luminophore cases (i.e., Dual, +FRS, +HPD, and +SOQ), photodegradation was substantially reduced under the argon atmosphere, indicating that the presence of oxygen affects photodegradation of the dual-luminophore PSP.

While FRS and HPD had almost no effect on photodegradation, the SOQ reduced the degradation rate of the tetranuclear Eu(III) complex by 31% compared with the original dual-luminophore PSP (i.e., Dual). This implies that singlet oxygen plays an essential role in the photodegradation mechanism of the dual-luminophore PSP. Because the temperature sensor does not produce singlet oxygen, the singlet oxygen molecules produced in the oxygen quenching reaction of PtTFPP are considered to react and destroy the tetranuclear Eu(III) complex molecules.

After the series of aging tests, additional sample coupons with the SOQ were prepared, and the effectiveness of the SOQ was further investigated. The formulation of dye solutions for additional coupons (hereby denoted as +2SOQ and +3SOQ) are listed in Table 4. One-hour-long aging tests were performed on those coupons.

The results of the additional aging tests are shown in Figure 6. The degradation rate of the tetranuclear Eu(III) complex was reduced by 91% in +2SOQ. However, the luminescence intensity was decreased by 40% and 95% in both wavelength ranges for +2SOQ and +3SOQ, respectively. Figure 7 shows the normalized luminescence intensity before the aging tests measured by the CCD camera at 100 kPa and 20 °C under the dry air atmosphere. The intensity was normalized by that of the dual-luminophore case (Dual) for each wavelength range. The concentration of the SOQ has a trade-off relationship between the degradation rate and the luminescence intensity. The SOQ used in this study absorbs UV and violet light [37,38], and the absorbance of excitation light is considered to cause a reduction in the luminescence intensity of the PSP.

### 4.2. Pressure and Temperature Sensitivities

The pressure and temperature sensitivities around 100 kPa and 20 °C before and after aging tests under dry air with and without the SOQ are listed in Table 5 and shown in Figure 8. Without antioxidants, the pressure sensitivity of the pressure sensor (PtTFPP) and the temperature sensitivity of the temperature sensor (tetranuclear Eu(III) complex) decreased after the aging test. The spectral overlap between the pressure and temperature sensors changes with photodegradation due to the difference in degradation rate. This is considered to cause a change in sensitivities. The sensitivities decreased with increasing the concentration of the SOQ. In contrast, the sensitivity changes after the aging tests were suppressed due to the mitigation of the spectral overlap change.

## 5. Conclusions

In the present study, three types of antioxidants—the free radical scavenger (FRS), the hydroperoxide decomposer (HPD), and the singlet oxygen quencher (SOQ)—were applied to the dual-luminophore PSP. The effects of antioxidants on the degradation rates and pressure/temperature sensitivities were investigated in one-hour-long aging tests with continuous excitation under dry air and argon atmospheres.

As a result of the aging tests, photodegradation was substantially reduced under the argon atmosphere, suggesting that presence of oxygen affects the photodegradation of the dual-luminophore PSP. A certain concentration of the SOQ reduced the degradation rate by 91% compared with the dual-luminophore PSP without antioxidants. This implies that singlet oxygen plays an essential role in the photodegradation mechanism of the dual-luminophore PSP. Because the tetranuclear Eu(III) complex does not produce singlet oxygen, the singlet oxygen molecules produced in the oxygen quenching reaction of PtTFPP are considered to react and destroy the tetranuclear Eu(III) complex molecules.

Whereas previous studies have explored photostable luminophore combinations empirically, the proposed method can suppress photodegradation and allows more flexible luminophore combinations with better characteristics (e.g., sensitivity and quantum efficiency). This leads to an accurate and quantitative surface pressure measurement.

## Figures and Tables

**Figure 1 sensors-22-09470-f001:**
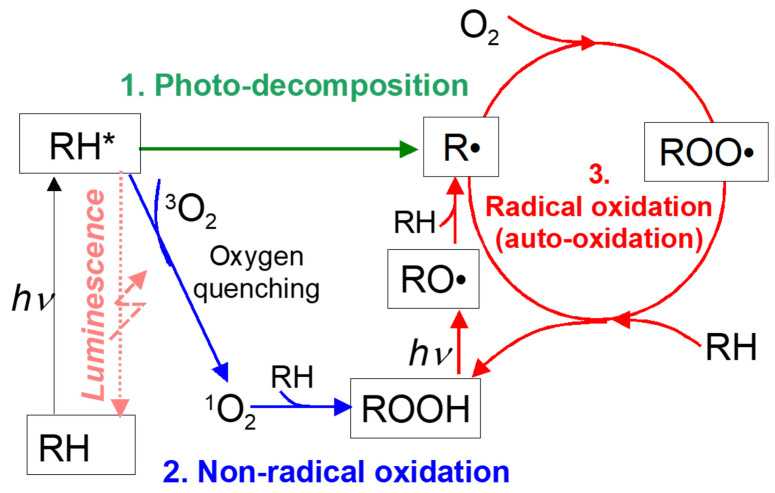
Schematic of photodegradation of an organic luminophore [33].

**Figure 2 sensors-22-09470-f002:**
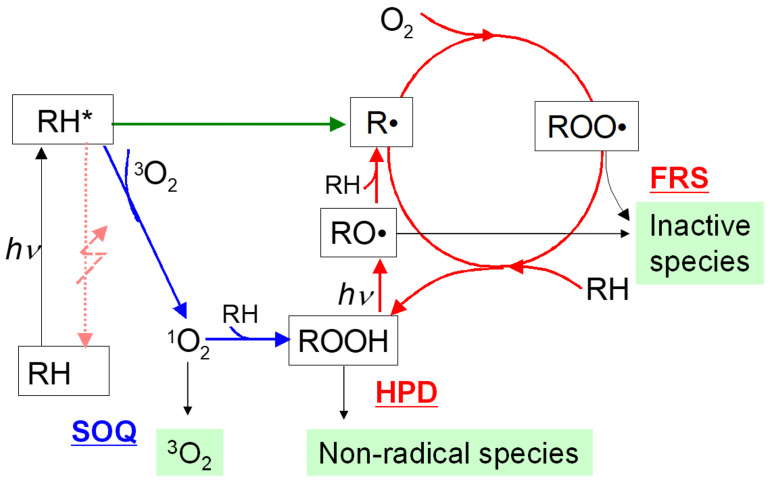
Schematic of photostabilization by antioxidants.

**Figure 3 sensors-22-09470-f003:**
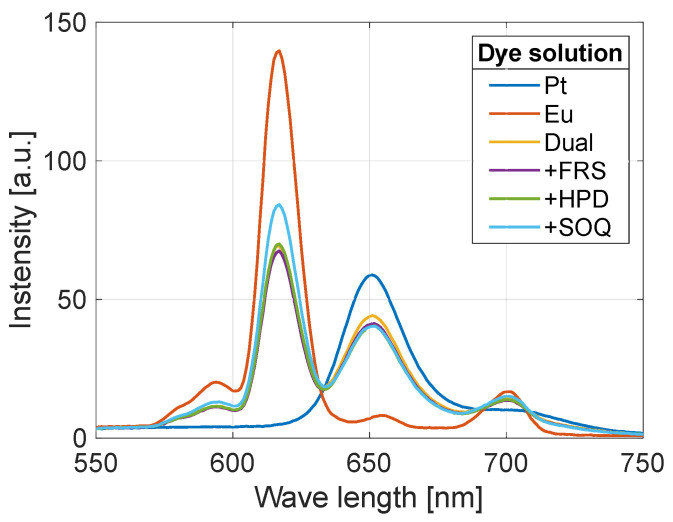
Luminescence spectra at 100 kPa and 20 °C (air).

**Figure 4 sensors-22-09470-f004:**
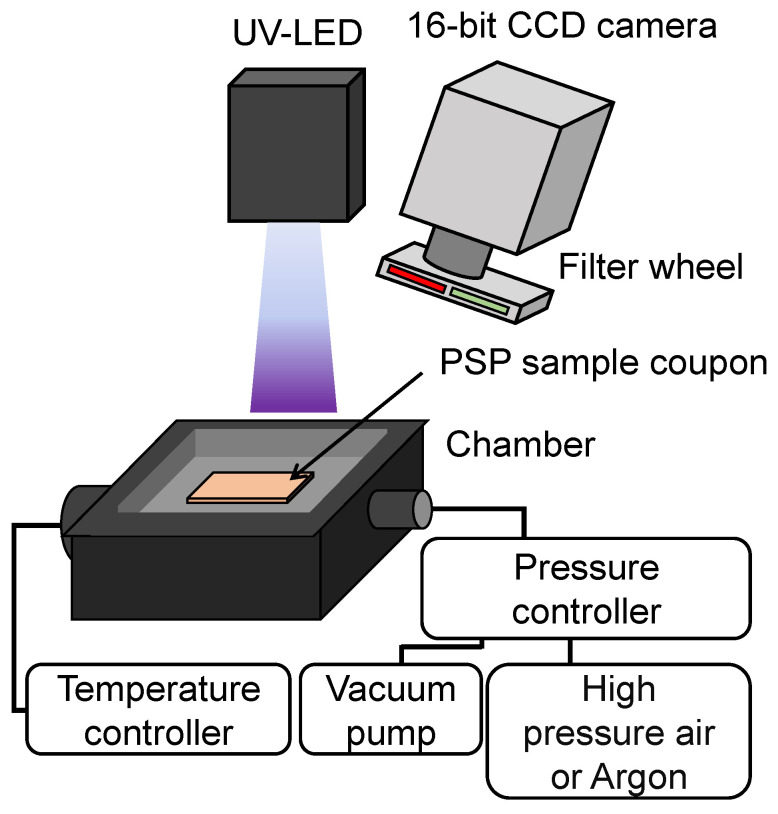
Schematic of the PSP calibration system.

**Figure 5 sensors-22-09470-f005:**
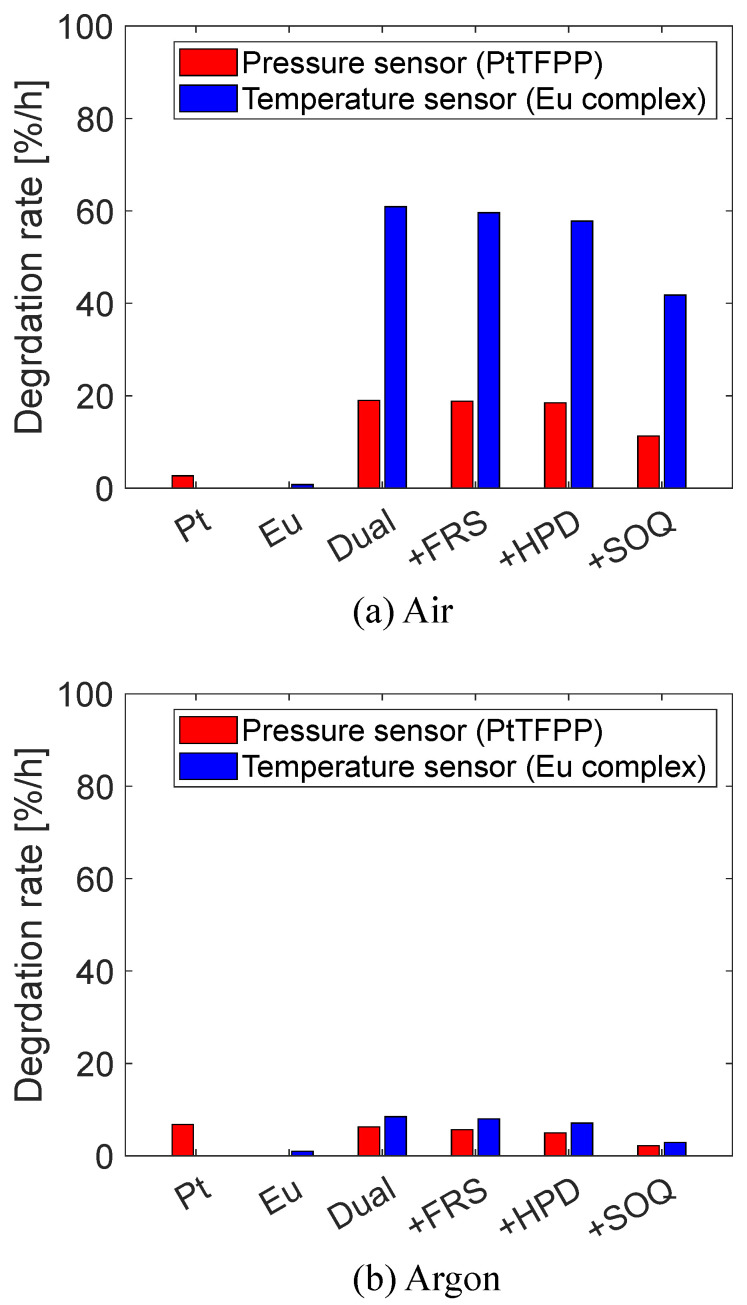
Degradation rates in the one-hour aging tests.

**Figure 6 sensors-22-09470-f006:**
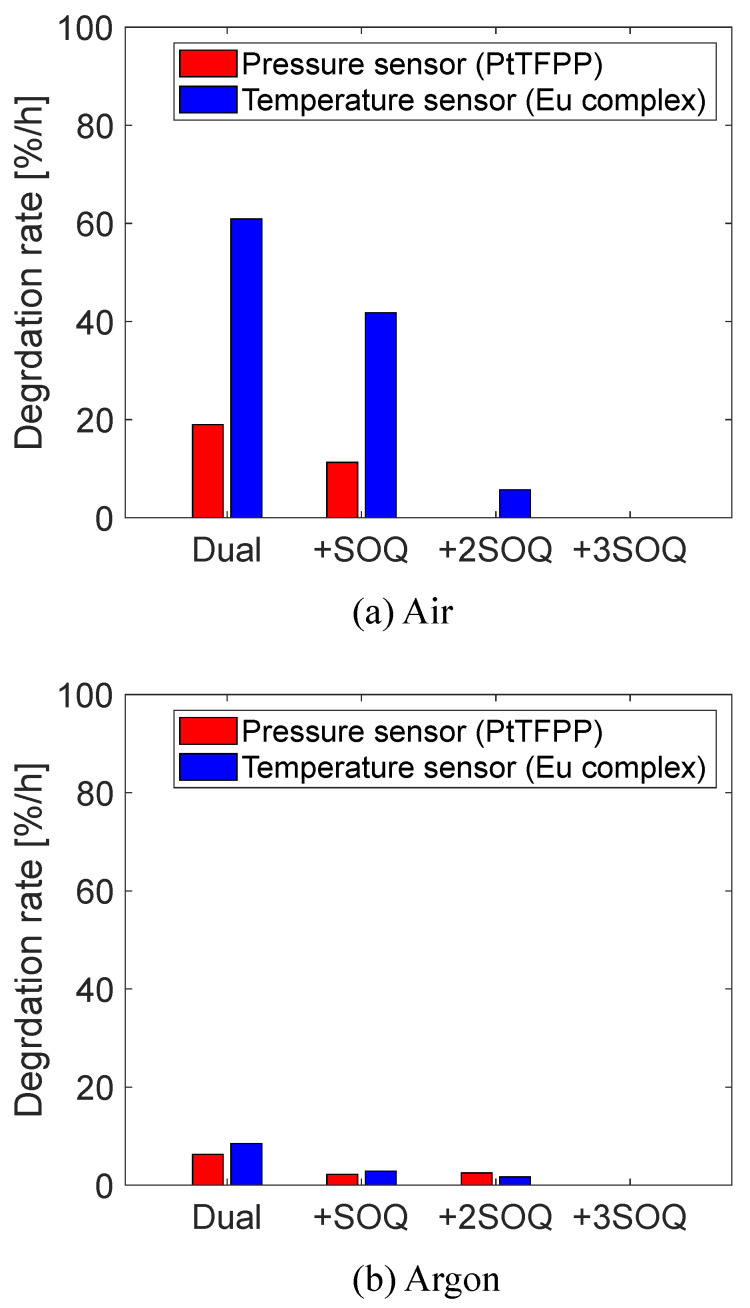
Effects of SOQ concentration on degradation rate.

**Figure 7 sensors-22-09470-f007:**
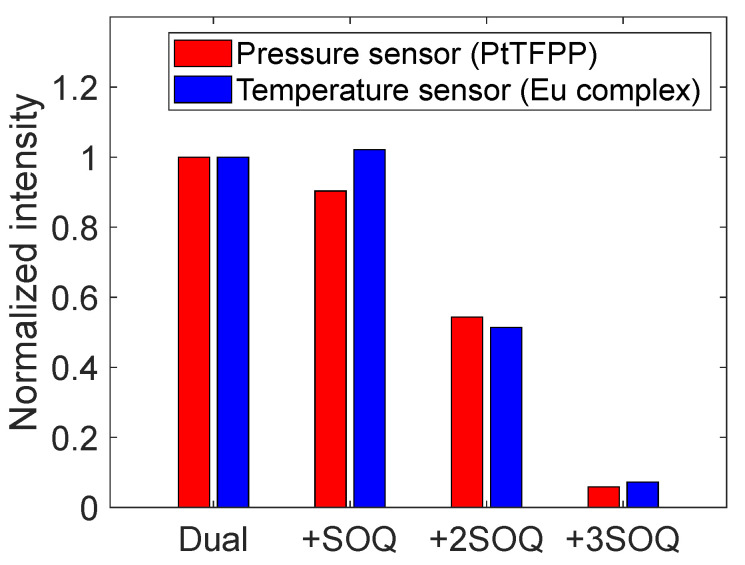
Effects of SOQ concentration on luminescence intensity.

**Figure 8 sensors-22-09470-f008:**
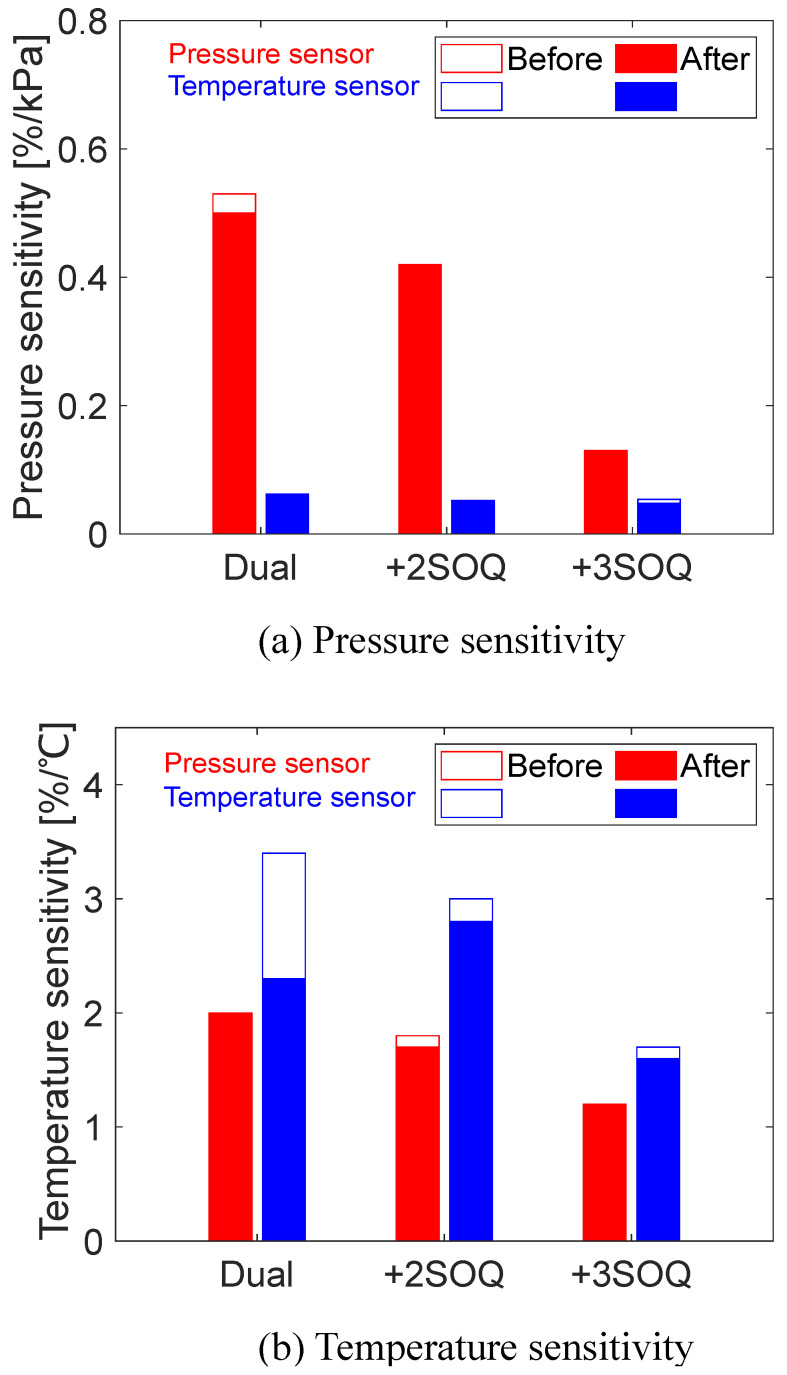
Pressure and temperature sensitivities before and after the aging tests.

**Table 1 sensors-22-09470-t001:** Binder solution.

Polymer	Particle	Solvent
Poly(IBM)	${\mathrm{TiO}}_{2}$	Toluene
0.500 g	6.64 mg	30 mL

**Table 2 sensors-22-09470-t002:** Dye solutions.

	Pressure Sensor PtTFPP	Temperature Sensor Tetranuclear Eu(III) Complex	Antioxidant	Solvent Toluene
Pt	4 mg	N/A	N/A	20 mL
Eu	N/A	41 mg	N/A	20 mL
Dual	4 mg	41 mg	N/A	20 mL
+FRS	4 mg	41 mg	FRS 0.4 mg	20 mL
+HPD	4mg	41 mg	HPD 0.4 mg	20 mL
+SOQ	4 mg	41 mg	SOQ 0.4 mg	20 mL

**Table 3 sensors-22-09470-t003:** Comparison of degradation rate with previous studies.

	Pressure Sensor	Temperature Sensor	Binder	Degradation Rate of Temperature Sensor
Present study	PtTFPP	tetranuclear Eu(III) complex	Polymer/ceramic [39]	46% for 30 min
Mitsuo et al. [40]	PdTFPP		Poly-IBM-co-TFEM	Ten times higher than alone
Mochizuki et al. [41]				55% for 30 min

**Table 4 sensors-22-09470-t004:** Dye solutions for additional tests.

	Pressure Sensor PtTFPP	Temperature Sensor Tetranuclear Eu(III) Complex	Antioxidant	Solvent Toluene
+2SOQ	4 mg	41 mg	SOQ 4 mg	20 mL
+3SOQ	4 mg	41 mg	SOQ 40 mg	20 mL

**Table 5 sensors-22-09470-t005:** Pressure and temperature sensitivities.

(a) Before Aging Tests				
	**Pressure Sensitivity [%/kPa]**		**Temperature Sensitivity [%/°C]**	
	**Pressure Sensor**	**Temperature Sensor**	**Pressure Sensor**	**Temperature Sensor**
Dual	0.53	0.023	2.0	3.4
+2SOQ	0.41	0.052	1.8	3.0
+3SOQ	0.13	0.054	1.2	1.7
**(b) After Aging Tests**				
	**Pressure Sensitivity [%/kPa]**		**Temperature Sensitivity [%/°C]**	
	**Pressure Sensor**	**Temperature Sensor**	**Pressure Sensor**	**Temperature Sensor**
Dual	0.50	0.062	2.0	2.3
+2SOQ	0.42	0.051	1.7	2.8
+3SOQ	0.13	0.048	1.2	1.6

## Data Availability

Data sharing not applicable.

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
