# Peer review of "Photostability Enhancement of Dual-Luminophore Pressure-Sensitive Paint by Adding Antioxidants"

_sensors, 2022, doi:10.3390/s22239470_

Round 1

Reviewer 1 Report

This paper describes a systematic study into the mechanisms of photodegradation and how they can be mitigated in the context of PSP testing. This is of gneuine importance to the community - especially using polymer ceramic paints.

The work is well described but the results can be presented in a more efficient way to aid comparison by the reader. Specifically Figures 7 and 8 can be combined with different shading patterns for pre and post aging results. 

The main mechanism of photodegradation (singlet oxygen radicalisation) is well highlighted for the luminophores involved. However, the main improvement of reducing this effect using SOQ has a significant drawback by reducing luminescent output. This is described in text but would be better shown using a repeat of Figure 3 for varying SOQ levels (the text reads as if the authors have completed this measurement). Although understanding of the mechanism is useful, the impact on paint output is of more direct importance to the PSP community. Also, as the spectral intensity was presented as arbitrary units, is there any consistency between the expected output?

Do the authors have a reference or example for the statement on line 34 about PSPs having high reflectivity to IR? The reference on the line previous [ref 17] states the emissivity of their PSP is approximately 0.91.

There are minor English language typos which need to be edited; however, nothing the obscures understanding, so the article would benefit from a final proof read.

Overall and enjoyable article showing the development of better understanding of the mechanisms of PSP degradation. 

Reviewer 2 Report

The paper is not in good format. The figures 7 and 8 present in references section. similarly figure 6 occurs in conclusion. It is difficult to check and review the manuscript in such unformatted and dispersed form.

Reviewer 3 Report

Dear Editor,

            This manuscript reports the Photostability Enhancement of Dual-Luminophore Pressure-Sensitive Paint by Adding Antioxidants

1.     Throughout the manuscript photo-stability should be replaced by photostability.

2.     Tables 2 and 3 (last column): soluvent should be replaced by cu solvent

3.     Compare results of this work with other reports (that were mentioned) in a table. please check.

4.     The text in lines 187-190 should be checked and completed according to figure 3.

5.     Figure 6 is not placed in the text where it is mentioned/discussed. This figure cannot be placed in the conclusion section.

6.     Table 4 should be placed in the text at once where it is mentioned.

7.     Figures 7 and 8 are not discussed in the discussion part of the paper. They cannot be placed between the references (they must be moved). The data in these figures (7 and 8) are repeated with those in Table 4. After removing the figures 7 and 8, three figures and a Table with the results of the present study remain.

8.     Minor points: Line 65: apressure sensor)= a pressure sensor); Line 198: was attached= were attached; Line 242: wavelengths ranges= wavelength ranges

Round 2

Reviewer 2 Report

Recommended for publication

Reviewer 3 Report

The authors have considered my comments. This paper could be accepted for publication